# Global warming accelerates soil heterotrophic respiration

Alon Nissan [1] ✉, Uria Alcolombri [1], Nadav Peleg [2], Nir Galili[3], Joaquin Jimenez-Martinez [1,4], Peter Molnar [1] & Markus Holzner[4,5]

Carbon efflux from soils is the largest terrestrial carbon source to the atmosphere, yet it is still one of the most uncertain fluxes in the Earth's carbon budget. A dominant component of this flux is heterotrophic respiration, influenced by several environmental factors, most notably soil temperature and moisture. Here, we develop a mechanistic model from micro to global scale to explore how changes in soil water content and temperature affect soil heterotrophic respiration. Simulations, laboratory measurements, and field observations validate the new approach. Estimates from the model show that heterotrophic respiration has been increasing since the 1980s at a rate of about 2% per decade globally. Using future projections of surface temperature and soil moisture, the model predicts a global increase of about 40% in heterotrophic respiration by the end of the century under the worst-case emission scenario, where the Arctic region is expected to experience a more than twofold increase, driven primarily by declining soil moisture rather than temperature increase.

Rising atmospheric carbon dioxide ($CO_2$) concentration is one of the primary contributors to global warming[1]. Within the terrestrial carbon cycle, soil respiration, the emission of $CO_2$ through root (autotrophic) and microbial (heterotrophic) respiration[2], is the largest carbon efflux into the atmosphere[1,3]. Therefore, reliable quantification of how soil respiration may be affected by climate change is critical for predicting future atmospheric $CO_2$ concentrations. However, estimating terrestrial carbon effluxes, primarily driven by soil respiration, is highly uncertain[4–7]. The global carbon budget is significantly impacted by terrestrial carbon fluxes, making it crucial to improve current estimates. Soil carbon fluxes are dependent on complex interactions between biological, chemical, and physical processes, which play out under fluctuating and heterogeneous environmental conditions. As a result, observing, measuring, and modeling soil carbon fluxes is challenging.

Soils play a vital role in transferring, buffering, filtering, and accumulating carbon at the interface between the atmosphere, biosphere, and lithosphere. For example, soils contain about three times as much carbon (1500–2400 PgC, 1 Pg = $10^{15}$ g) as the atmosphere (600–800 PgC) or the Earth's vegetation (450–650 PgC)[1,8,9]. Roughly a fifth of atmospheric $CO_2$ originates from soils (~110 PgC $yr^{-1}$), which is about ten times more than anthropogenic $CO_2$ emissions (~11 PgC $yr^{-1}$)[1]. Soil heterotrophic respiration (HR) is one of the primary mechanisms through which terrestrial ecosystems release $CO_2$ into the atmosphere, and its relative contribution has been observed to gradually increase over the past two decades[10]. HR varies over a wide range of time scales (e.g., daily fluctuations and seasonal cycles), and is principally controlled by two climatic variables: soil temperature and moisture[11]. While soil temperature is positively correlated with HR[12,13], soil moisture shows a non-monotonic relationship[14]. Low soil moisture content reduces HR rates by limiting solute flux due to poor water connectivity in the pores. High moisture content reduces HR by limiting oxygen ($O_2$) supply from the atmosphere due to the weak

[1]Institute of Environmental Engineering, Department of Civil, Environmental and Geomatic Engineering, ETH Zürich, Zürich 8093, Switzerland. [2]Institute of Earth Surface Dynamics, University of Lausanne, Lausanne 1015, Switzerland. [3]Geological Institute, Department of Earth Sciences, ETH Zürich, Zürich 8092, Switzerland. [4]Department of Water Resources and Drinking Water, Swiss Federal Institute of Aquatic Science and Technology, EAWAG, Dübendorf 8600, Switzerland. [5]Biodiversity and Conservation Biology, Swiss Federal Institute for Forest Snow and Landscape Research, WSL, Birmensdorf 8903, Switzerland. ✉e-mail: anissan@ethz.ch

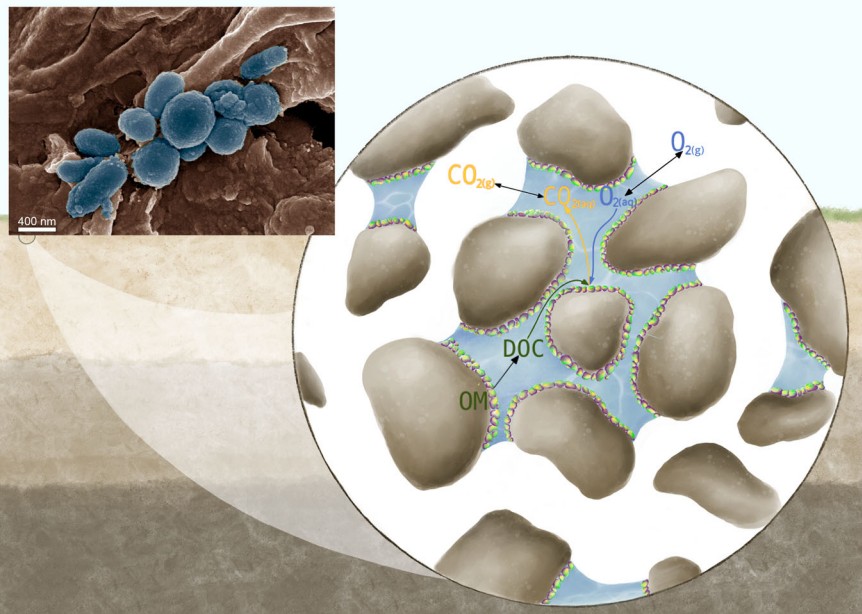

**Fig. 1 | Schematic of the reactive transport model at the pore-scale.** $O_{2(g)}$ diffuses from the atmosphere (gas phase) and is dissolved to the liquid phase ($O_{2(aq)}$) according to Henry's law. Simultaneously, dissolved organic carbon (DOC) is released from soil organic matter (OM). In the presence of both $O_{2(aq)}$ and DOC, aerobic heterotrophic respiration occurs at the interface between the aqueous phase (blue) and the surface of soil grains (brown) by attached microbes (biofilms). As a result, $CO_{2(aq)}$ is released to the water and to the atmosphere $CO_{2(g)}$. In the inset, a scanning electron microscope image shows a cluster of bacteria (bluish colored) around a micrometer root (brown); false coloring was applied for illustration purposes.

diffusivity of $O_2$ in water relative to gas. As a result, the response of HR to soil water saturation shows a bell-shaped response curve, with optimal conditions for respiration at intermediate soil moisture content[15].

Modelling HR rates based on soil temperature and moisture is challenging due to the many interacting processes that are controlled by these variables[11,16]. Consequently, most modelling efforts to quantify changes in HR rates as a function of temperature and moisture employ empirical parameterizations using macro-scale (bulk) properties of the soil (e.g.,[15,17]) or empirical fits using a variety of function shapes (e.g.,[16,18,19]). These parameterizations often have no direct connection to biophysical processes, and are site dependent rather than universal laws[16].

Here, we provide a mechanistic perspective on soil HR built upon pore-scale processes at the grain interface where microbial communities reside (see inset of Fig. 1 as an example) and link this to global-scale patterns and future trends. We first quantified soil HR starting from the pore (micro) scale, integrating parameters that are biological (e.g., microbial respiration and production of $CO_2$), chemical (reaction processes), and physical (transport mechanisms and soil texture). Subsequently, we upscaled the pore-scale HR relations by defining scaling laws from pores to water patches using percolation theory[20], which allows us to make predictions of HR fluxes for larger (field) scales while maintaining the biophysical representation of the pore scale. Then, considering soil temperature and moisture variation in space and time, we show that the model yields estimates of recent trends in soil HR rates at the global scale that are in line with observations. Finally, we use this mechanistic model to simulate how soil HR might change under the worst-case future climate scenario from CMIP6 climate change experiments.

## Results

### Soil heterotrophic respiration at the pore scale

To quantify the non-linear relationship between HR and soil moisture content[14], which is considered to be the most uncertain parameter when estimating soil HR[16], we started by performing pore-scale

numerical simulations using an image-based percolation algorithm, to obtain the air–water distribution in the soil matrix under different saturation degrees[21] (Methods and Supplementary Fig. 1a). Different soil configurations were examined by changing the characteristic grain diameter $\lambda_c$. The soil configurations were generated using a random distribution of circular grains with radius $\lambda_c$.

Based on the air–water distribution in the soil from the percolation simulations (Supplementary Fig. 1b), reactive transport simulations were computed on the 2D pore-scale domains[22] (Methods and Fig. 1) to quantify the response of HR to soil saturation degree ($\tilde{\theta} = \theta/\theta_s$, where $\theta_s$ is the moisture capacity of the porous media). In these simulations, other environmental conditions (e.g., ambient temperature and reaction parameters) were kept constant (Supplementary Table 2). The inset in Fig. 2a shows the model output, revealing the expected non-monotonic relationship between $\tilde{\theta}$ and mean soil HR, $\tilde{R}_h = \int_\Omega \frac{R_h}{V_m} d\Omega$, where $R_h$ is the local respiration [mol m$^{-2}$ s$^{-1}$], $V_m$ is the maximum respiration rate [mol m$^{-2}$ s$^{-1}$] from the surface of grains, and $\Omega$ is the length of the solid–liquid interface [m]. As expected, $\tilde{R}_h$ increases as the characteristic grain size ($\lambda_c$) decreases, due to the increase in the surface area.

To make the connection between pores and continuum (bulk) scale processes, we examine the flux $\tilde{R}_h$ as a function of the number of water patches[23] in the domain $N_c$ and the mean characteristic size of those water patches $S_c$, both dependent on saturation degree $\tilde{\theta}$. Simulations show that there is a characteristic scaling relation $\tilde{R}_h/N_c \sim S_c^{0.5}$, which indicates a proportionality between the respiration rate within a single water patch and its size (dots in Fig. 2a).

From the scaling laws of percolation theory[20], we derive the characteristic number ($N_c$) and sizes ($S_c$) of water patches within the domain[23], for different water saturation degrees (see Methods). The resulting theoretical relations of $N_c$ and $S_c$ to $\tilde{\theta}$ at the continuum scale (Eqs. (2), (3) in Methods) were compared successfully against the results from numerical percolation simulations, and are shown in Supplementary Fig. 2a, b. In addition, to further validate Eqs. (2) and (3), we conducted laboratory drainage experiments using microfluidic chips with different grain sizes $\lambda_c$, in which we compared simulated and

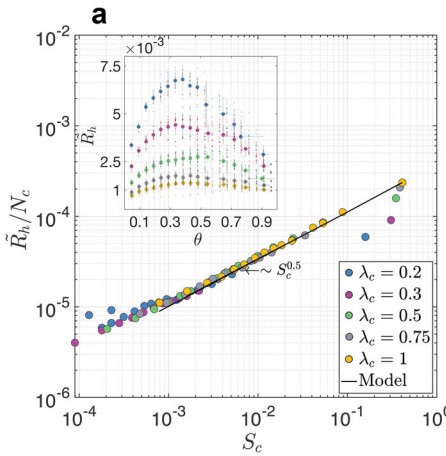

**Fig. 2 | Heterotrophic respiration from pore to continuous scale. a** Simulated mean dimensionless heterotrophic respiration rate per water patch ($\tilde{R}_h/N_c$) as a function of the water patch characteristic size ($S_c$). The inset shows the non-monotonic relationship between the mean soil HR ($\tilde{R}_h$) and soil saturation ($\tilde{\theta}$), for different grain sizes $\lambda_c$ (in mm units). The error bars represent the standard deviation of 20 realizations. **b** Dimensionless heterotrophic respiration at the scale of a single water patch as a function of $\beta$, for different $\alpha$ values (Methods, Eq. (7)). The solid lines represent the analytical solutions (Methods) for the limiting cases; $u$ is the dimensionless substrate concentration. Insets show the spatial distribution of the reaction within a water patch ($\alpha = 0.1$), for $\beta = 0.1$ (left inset) and $\beta = 1000$ (right inset). The colour bar represents the dimensionless respiration rate in logarithmic scale.

observed water patch properties (Methods and Supplementary Fig. 2c, d).

## Soil heterotrophic respiration at the patch scale

Following the results of pore-scale simulations in Fig. 2a, which demonstrate the basic relation between water patch size ($S_c$) and respiration rate, we can now generalize the problem to account for variation in environmental conditions (i.e., soil texture, temperature, substrate concentration, diffusion and reaction parameters) by formulating a steady-state Diffusion Reaction Equation (DRE) at the scale of a single water patch. The mathematical formulation is based on three general assumptions: (i) the reactive volume can be treated as homogeneous within the water patch and proportional to the grain surface area, (ii) the substrate (dissolved organic carbon) concentration is uniformly distributed within the water patch, and (iii) at steady state conditions, microbial activity is proportional to the substrate fluxes.

In non-dimensional form, with symmetrical and spherical coordinates, the DRE can be written as (see Methods for the mathematical development),

$$\frac{d^2u}{d\chi^2} + \frac{2}{\chi}\frac{du}{d\chi} = \frac{\beta u}{u+\alpha}, \tag{1}$$

where $u$ is the normalized oxygen concentration, $\beta$ and $\alpha$ account for the biophysical parameters of the soil, solute, reaction, and characteristic water patch (Methods Eq. (6)), and $\chi$ is the normalized spatial coordinate. An analytical solution for Eq. (1) is not available, except for limiting cases ($\alpha << u$ or $\alpha >> u$). We can, however, solve Eq. (1) numerically, to obtain the (non-dimensional) total respiration rate as a function of $\beta$ for example, for different $\alpha$ values for a characteristic water patch (dots in Fig. 2b). To examine the validity of Eq. (1), we compare it with the results from the 2D numerical reactive transport simulations (Fig. 2a). The solid black line in Fig. 2a represents the numerical solution of Eq. (1) for $\lambda_c = 1$ mm.

Numerical solution of Eq. (1) reveals the effect of $\beta$ on HR at water patch scale for different $\alpha$ values (dots in Fig. 2b). High $\beta$ values are associated with large water patches (i.e., high $\tilde{\theta}$), porous media with high surface area (i.e, small $\lambda_c$), small diffusion coefficient (e.g., at low temperature) and high reaction rates. As a consequence, at high $\beta$, the local reaction rate within a water patch shows spatial variability, where dissolved $O_2$ is rapidly consumed at the surface of the patch and does

not penetrate into the patch's interior (Fig. 2b right inset, $\beta = 1000$). On the other hand, at low $\beta$ values, reaction production is uniform in space (Fig. 2b left inset, $\beta = 0.1$), and there is no $O_2$ limitation within the water patch. In general, as the values of $\alpha$ decrease and $\beta$ increase, more respiration (i.e., $CO_2$ efflux) takes place in the water patch; for more details see Methods.

To test our modelling framework, we compared the simulated and observed HR rates with published laboratory and field measurements demonstrating the dependence on soil saturation and temperature[24–28]. To define the soil parameters for these comparisons, the reported soil texture was used to estimate $\lambda_c$[29] and porosity $\phi$. Other parameters (e.g., diffusion coefficient, oxygen concentration) were derived using the temperature conditions, while the substrate (DOC) concentration was assumed saturated with respect to Michaelis–Menten kinetics. Despite the considerable experimental scatter and the uncertainties inherent in initializing some of the model parameters, the observations and model outputs were in good agreement (Supplementary Fig. 3).

## Soil heterotrophic respiration at global scale

To validate the predictions from our model for various climatic locations, we compare the results with the global soil respiration database[30] (Supplementary Fig. 3d). To estimate the model parameters (Eqs. (6), (7)), we used global databases (at a spatial resolution of 0.25° and at monthly intervals) to characterize the soil temperature and saturation degree[31], soil texture[32], and dissolved organic carbon concentration[33] (for more details see Methods and Supplementary Fig. 6). We computed soil HR at monthly resolution, and aggregated the values to obtain a mean annual HR representing the time period of each observation. Despite large uncertainties in the global climatic and soil databases, the results from our model are in good agreement (RMSE = 214 gC m$^{-2}$ yr$^{-1}$).

The globally simulated top soil (0–10 cm depth) annual HR is presented in Fig. 3a. We find that the average global HR rate from the topsoil layer is -282 gC m$^{-2}$ yr$^{-1}$, which is consistent with previous estimates[6,34,35] and represents approximately 70% of current estimates of the total global soil HR (about 42 PgC yr$^{-1}$). This highlights the dominant contribution of the topsoil to soil HR resulting from the fact that this layer is usually under semi-saturated water conditions, at a higher temperature than deeper soil layers, rich in soil organic material, and abundant heterotrophic organisms.

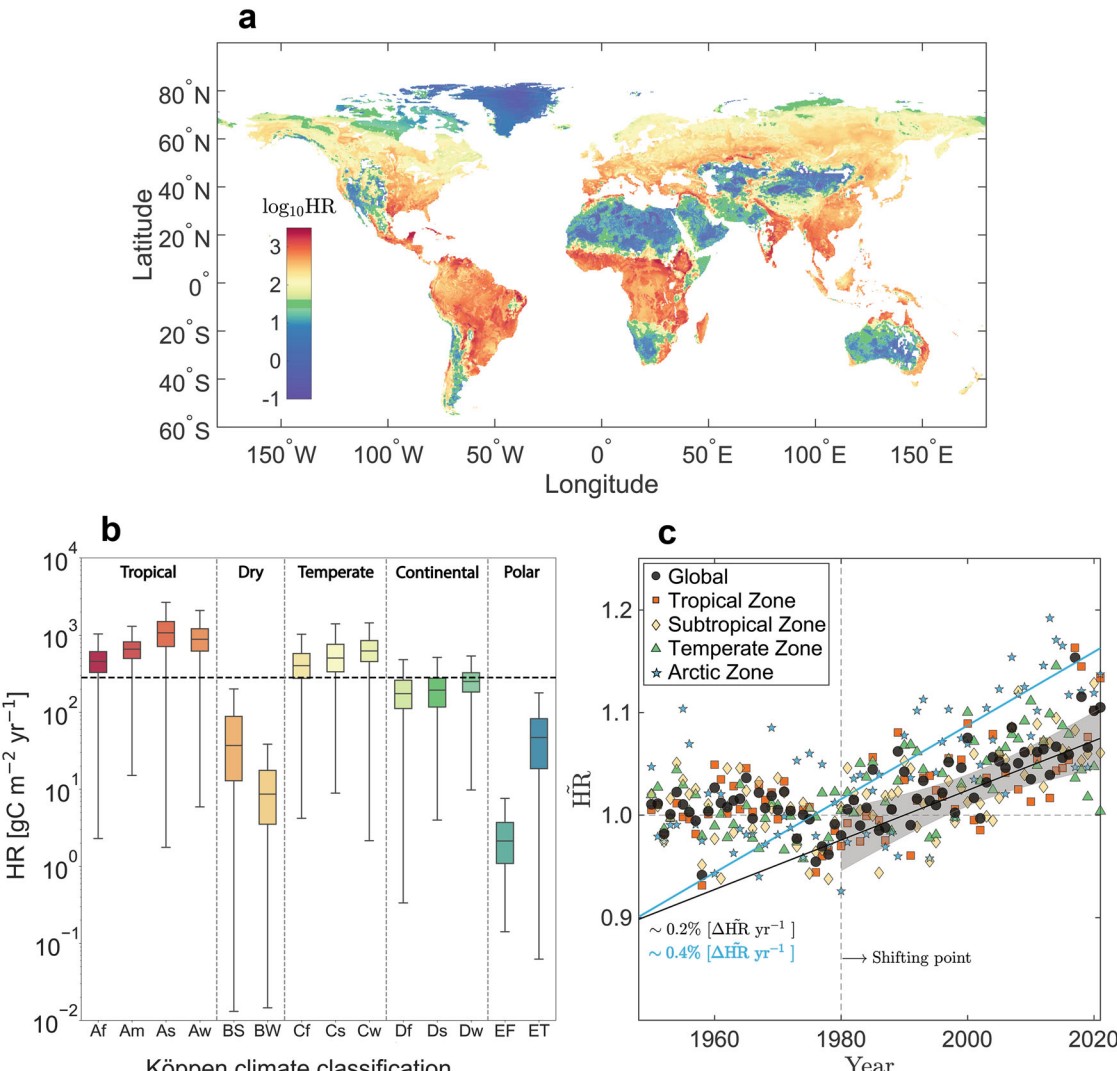

**Fig. 3 | Heterotrophic respiration at the global scale. a** Global map of simulated mean annual soil heterotrophic respiration (HR) rate (for year 2021); colour bar in logarithmic scale ($\log_{10}$(HR), where HR units are gC m$^{-2}$ yr$^{-1}$). White regions represent aquatic environments or missing data. **b** HR is shown for various Köppen climate classifications. **c** Temporal trend in soil HR (normalized by the mean HR value from 1948 to 1980) over the period 1948–2021; colours represent different latitudinal climatic regions. The black line is a linear fit to the global data (with shading indicating the 95% confidence interval for the fit), and the cyan line is the fit for the Arctic zone.

To evaluate the impact of input uncertainties in the monthly soil temperature and moisture data on our model predictions, we conducted a sensitivity analysis using a daily database[31]. The analysis involved 2000 Monte Carlo simulations, in which representative monthly soil temperature and moisture values were generated by randomly selecting from their daily distributions at each grid cell. The results of the simulations were analyzed to determine the annual HR estimate and its level of variability. As shown in Supplementary Fig. 5a, the highest level of sensitivity was estimated at 5% and found at the southern pole ($\pm 12$ gC m$^{-2}$ yr$^{-1}$), while the highest standard deviation of about $\pm 20$ gC m$^{-2}$ yr$^{-1}$ was found at the tropics. The robustness of the model is confirmed by these results, which are further discussed in the Methods section.

**Present and future trends in soil heterotrophic respiration**

In light of current trends in surface temperature and soil moisture[1,36], and the fact that these two are the main factors controlling soil respiration[11], we estimate the temporal evolution of soil HR over the last 73 years (1948–2021) based on these two climatic variables from the Global Land Data Assimilation System database[31] (Fig. 3c). Data for

each geographical zone and the global value are normalized by the mean soil HR from 1948 to 1980. A statistically significant trend ($p < 0.01$; Mann–Kendall test) is seen from 1980 onwards, where soil HR shows an average global increase of $0.2\% \pm 0.05\%$ each year. This is equivalent to an average increase of around 6 gC m$^{-2}$ per decade. Similar trends are found for all geographical zones except the Arctic, where model output suggests a greater increase of $\sim 0.4\% \pm 0.11\%$ annually. These findings are in agreement with field observations for 1989–2008[30], remote sensing data for 2000–2014[37] of the overall soil respiration, and data-driven models[19,38,39].

The global trend estimated by our model is roughly 1.5 times higher than previous estimates computed using a machine learning algorithm[38] based on the SRDB database[30], with low temporal resolution and for the entire soil column. The topsoil layer is an interface to the atmosphere and thus is more sensitive to environmental feedback and changes. Hence, the discrepancy between the two models, which use different approaches (mechanistic and data-driven) and soil layers, falls within reasonable limits. In the Arctic zone, although the change in HR is already meaningfully higher than in the other geographical regions ($\sim \times 2$), soil HR shows highly scattered values (Fig. 3c). This is

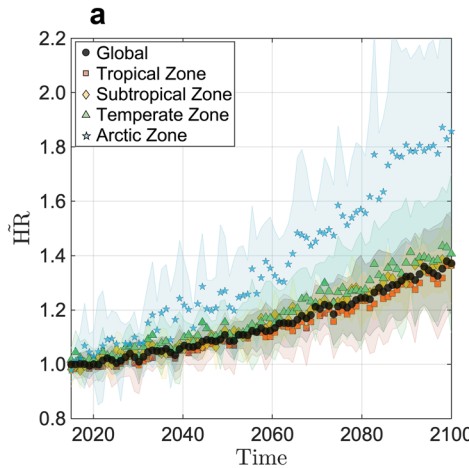

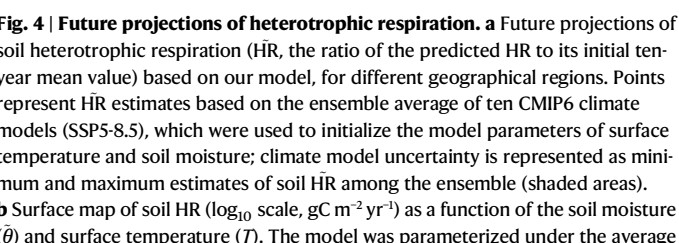

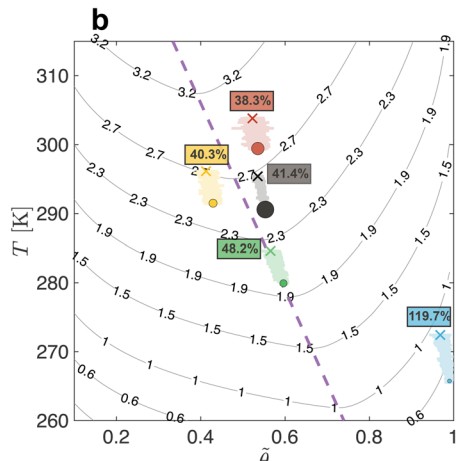

**Fig. 4 | Future projections of heterotrophic respiration. a** Future projections of soil heterotrophic respiration (H̃R, the ratio of the predicted HR to its initial ten-year mean value) based on our model, for different geographical regions. Points represent H̃R estimates based on the ensemble average of ten CMIP6 climate models (SSP5-8.5), which were used to initialize the model parameters of surface temperature and soil moisture; climate model uncertainty is represented as minimum and maximum estimates of soil H̃R among the ensemble (shaded areas). **b** Surface map of soil HR ($\log_{10}$ scale, gC m$^{-2}$ yr$^{-1}$) as a function of the soil moisture ($\tilde{\theta}$) and surface temperature ($T$). The model was parameterized under the average

global conditions of soil grain size and substrate concentration. The large point for each geographical zone corresponds to its current state in 2021, with symbol size representing the relative contribution to global soil HR (Tropical 67%, Subtropical 23%, Temperate 10% and Arctic 0.1%). Projected annual changes in soil moisture and surface temperature are indicated by small points; the variance between the climatic models is represented by the error-bars along each trajectory. The symbol **X** at the end of each trajectory represents the predicted variable values at the end of the century. The dashed purple line indicates the ridge of the HR surface.

due to extreme fluctuations between seasons in the Arctic; as an example, in 2010, due to the extremely hot summer[40], the model indicates a 15% increase in soil HR compared to the baseline. As these fluctuations have become more frequent in the last decade (Fig. 3c, 2010 onwards), soil HR in the Arctic might increase considerably faster than 0.4% per year.

Finally, to predict potential future trends, we used data from 10 Global Circulation Models (Coupled Model Intercomparison Project phase 6, CMIP6[41], resolution of 1° and monthly) to assess how changes in surface temperature and soil moisture might affect the evolution of soil HR until the end of the century (Fig. 4a). We employ the worst-case greenhouse gas emission scenario (SSP5-8.5 scenario) as an upper limit for future projections of soil HR. On this basis, the model predicts a dramatic increase in soil HR emission in all geographical zones (~40%), with a much greater increase in the Arctic zone (~100% increase by 2100).

Recently, Lynch et al.[42] carried out a comprehensive evaluation of 25 Earth System Models (ESMs) under the SSP5-8.5 scenario. The findings reported in their paper exhibit significant disparities and uncertainties among the models. The mean projection of HR by the end of the century is a 50% increase, while only a few ESMs successfully reproduce the historical HR. Our model provides a lower estimate of HR increase while retaining consistency with prior observations.

To better understand the mechanisms underlying the predicted HR trends, we computed and displayed the projected mean soil HR as a function of the mean surface temperature and soil saturation (Fig. 4b). All geographical zones are predicted to experience a strong increase in surface temperature and a moderate decrease in saturation (Fig. 4b); this is predicted to result in a strong increase in the soil HR rate (percentage values within Fig. 4b). The increase of surface temperature and decrease of soil moisture cause the soil HR to increase in parallel with the surface maximal gradient, which leads to a (near) maximal rate-of-change in the tropical and temperate zones − and also globally. Notably, some of the geographical zones are in the vicinity of the maximum HR potential (purple dashed line) based on their mean temperature and soil moisture (most clearly the tropical and temperate zones). This might suggest an adjustment of the soil microbial activity to maximum production in these zones. Changes in soil HR in

all geographical zones except the Arctic show an increase of 36–42% by the end of the century (Fig. 4b). The Arctic zone occupies a unique place on this surface. While current observations reveal a considerable increase in surface temperature but little change in soil moisture, climate models project instead a future decrease in soil moisture, which results in a sharp increase in the predicted soil HR by > 100%. This strong increase stems from the curvature of the HR surface at the Arctic location, where the maximum change is oriented upwards and to the left on the surface. The model response surface indicates that a reduction in soil moisture, in all zones except in the Arctic, will lead to a reduction in soil HR.

## Discussion
We have presented a biophysical model to estimate soil heterotrophic respiration from climatic data. The principal innovation of our approach is the mechanistic perspective on soil HR from pore to global scales, in the absence of any empirical parameters, which enables robust quantification of current HR rates and future trends. The simplicity of the model is encapsulated by the two variables, $\alpha$ and $\beta$ (Methods Eq. (6)), that control the respiration rate at the scale of a single water patch (Fig. 2b). Moreover, as the value of $\alpha$ is strictly limited ($0.01 < \alpha < 0.5$) for aerobic respiration, it is mainly $\beta$ that controls the reaction rate within a single patch (Supplementary Fig. 4a, b). We validated the model against controlled laboratory measurements and field observations, and find a good agreement with the predictions (Supplementary Fig. 3).

A fundamental assumption of the model is that the ambient conditions control microbial functioning[43], despite the taxonomic diversity of microbes on Earth[44,45]. This assumption is supported by the relative importance of $\beta$ over $\alpha$ (Fig. 2b) in determining HR rate at the single patch level, as $\beta$ characterizes the environmental conditions while $\alpha$ accounts for microbial activity.

As demonstrated by the global-scale analysis (Fig. 3c), HR from the topsoil layer is increasing globally at a rate of about 2% per decade (equivalent to an increase of about 0.7 PgC per decade). This trend is similar in all geographical locations except for the Arctic zone, where the rate is double the global mean (about 4% per decade). Based on projections of changes in soil temperature and moisture obtained

from climate models, we demonstrated that future trends indicate a gradual increase in annual soil HR rate until the end of the century (5% per decade), with a greater increase (10% per decade) in the Arctic zone (Fig. 4a). Within the Arctic zone, unlike other climatic zones, the change in soil HR is principally driven by a projected reduction in soil moisture rather than by the increase in temperature (Fig. 4b). According to current observations, the soil moisture content in the Arctic does not yet show a negative trend and remains on average close to saturation. The increase in soil respiration in the Arctic over the last four decades (Fig. 3c) is thus still mainly due to increased surface temperature, and the onset of declining soil moisture may represent a tipping point.

Our model predictions, as depicted in Figs. 3, 4, treat DOC as a static or quasi-static input variable[46], rather than considering it as a dynamic outcome of soil carbon fluxes[47]. To test our hypothesis that soil temperature and water content are the primary factors controlling the availability of DOC for HR, we simulated two extreme scenarios of DOC concentration until 2100 using predictions from Earth System Models (Todd–Brown et al., 2014): a 20% linear increase and a 5% linear decrease. The results, depicted in Supplementary Fig. 5b, demonstrate that soil temperature and moisture are the primary regulators of DOC availability for microorganisms, even under these two end-member scenarios. Our modeled HR trends are consistent with observations[6,34,35], highlighting the significant role of soil temperature and moisture as key determinants of HR efflux and, consequently, DOC availability. A more explicit inclusion of DOC as a dynamic input variable into our model could be relevant for predicting HR in ecosystems where DOC fluctuates at short temporal scales. However, this would require consideration of dependencies of DOC on several factors, including vegetation, root exudates, microbial activity, precipitation, temperature, and moisture, and is beyond the scope of the present work.

Land-use changes, such as deforestation[48], reforestation[49], afforestation[50], and changes in agriculture and land management[51–53], can have a significant impact on the terrestrial carbon cycle by affecting carbon storage in soil and vegetation. These effects can be both positive or negative, and their future predictions are highly uncertain[1]. While our study does not consider the effects of these changes, the model can be parameterized to reflect future changes in land use by modifying the soil and dissolved organic carbon parameters. It should be noted that current estimates indicate that land-use changes contribute about 0.1 PgC per year to the atmosphere, while HR estimates are around 50–60 PgC per year[1]. Hence, even without considering land-use changes in the model's future projections, the results are still valid given their relatively minor contribution.

At large time scales, carbon efflux cannot exceed the carbon influx by Gross Primary Production. The ratio between HR and autotrophic respiration has increased over the past three decades[10]. Those two facts, together with the predicted HR increase, are a manifestation of the non-equilibrium conditions of the terrestrial ecosystem, where HR, at short time scales (i.e., up to hundreds of years), is independent of Gross Primary Production, eventually leading to a loss of the soil carbon stocks[54,55]. How future changes in the Earth's climatic environment will affect the terrestrial carbon cycle is one of the primary concerns of the 21st century, and this study sheds light on one of its primary mechanisms.

## Methods

### Image-based invasion percolation algorithm
An image-based invasion percolation algorithm[21,56] was used to simulate the water spatial distribution at different saturation degrees ($\bar{\theta}$) for different porous medium configurations. The porous materials were generated using randomly distributed grains of uniform diameter ($\lambda_c$) within the domain (20 realizations for each $\lambda_c$); each porous medium system had overall dimensions of 3 cm (width) × 2.25 cm (height), and a porosity of 0.5. Initially, the domain is filled with water, and during the invasion percolation process, the air is entering from the upper boundary of the system and invades the liquid phase, following the capillary entry conditions at the air–water interface. At the end of the simulations, the percolation clusters (Supplementary Fig. 1a) and thus the distribution of water patches within the domain (Supplementary Fig. 1b) is obtained at different saturation degrees. Note that even though we used a uniform grain size for each realization, the resulting simulated porous media are highly heterogeneous in their pore size distributions (see inset in Supplementary Fig. 1b as an example). For more details see[56].

### Estimating water patch proportionalities from percolation theory.
From percolation theory, we expect that the size distribution of water patches $n(s)$ in the domain follows a general proportionality[20], $n(s) = s^{-\tau}\exp(-s/\xi)$, where $s$ is the water patch size, $\tau$ is the Fisher exponent, which depends on the dimensionality of the system, and $\xi$ is a cut-off parameter, which depends on the porous medium properties and the saturation degree[23]. From this definition of $n(s)$, we calculate the characteristic number of patches in the domain using the mean of the distribution function, $N_c = \int_0^1 n(\bar{s})d\bar{s}$ (where $\bar{s} = s/\theta_s$), which yields $N_c = \left(\frac{1}{\xi}\right)^{\tau-1}\Gamma(1-\tau) - E_\tau\left(\frac{1}{\xi}\right)$, where $\Gamma$ is the gamma function and $E_\tau$ is the exponential integral function. From mass conservation, the characteristic patch size is

$$S_c = \bar{\theta}/N_c. \qquad (2)$$

From percolation simulations, the value of $\xi$ is found to depend on $\bar{\theta}$ as $\xi = \gamma(1-\bar{\theta})$, with $\gamma = 100$. Finally, to account for the total number of water patches in the domain of length scale $L$, $N_c$ is normalized by $N^0 = \left(\frac{L}{\lambda_c}\right)^{d(1-\bar{\theta})}$[23], to yield

$$N_c(\bar{\theta},\lambda_c) \approx N^0(\bar{\theta},\lambda_c)\left(\frac{1}{\xi(\bar{\theta})}\right)^{\tau-1}\Gamma(1-\tau) - E_\tau\left(\frac{1}{\xi(\bar{\theta})}\right), \qquad (3)$$

where $d$ is the dimensionality ($d = 2$ for the numerical simulations). According to this formulation, under fully saturated conditions ($\bar{\theta} = 1$), $N_c = 1$ (thus a single water cluster) and the characteristic patch $S_c$ has the size of the entire system voids. The resulting theoretical relations of $N_c$ and $S_c$ to $\bar{\theta}$ at the continuum scale (Eqs. (2)–(3)) are shown in Supplementary Fig. 2a and b, respectively, and compared to the numerical percolation simulations.

### Microfluidic experiments
To support the relationships between water patch properties and soil saturation degree (Eqs. (2),(3)) obtained from percolation theory, we performed microfluidic experiments. Microfluidic chips were fabricated using soft lithography[57]. Each chip had overall dimensions 5 cm × 5 cm × 0.005 cm. Similar to the numerical simulations (Methods), the solid phase was generated by randomly distributing circular objects ("grains") with diameter $\lambda_c$ mm. To mimic the invasion percolation algorithm, the microfluidic chips were placed with a long axis vertical and saturated with water (dyed with fluorescein solution of 0.01 mM) as initial condition. The upper side of each chip was open to the atmosphere, and the lower side was open for drainage. The air and water phases were monitored during the drainage process using a CCD camera (Ximea, Germany), and the relationship between the number and size of water patches and the saturation degree was obtained by image analysis (Supplementary Fig. 2).

The divergence of the experimental data (Supplementary Fig. 2) from theory observed at very low saturation conditions is expected (as $\bar{\theta} \to 0$, also $N_c \to 0$), because the relations in Eqs. (2)–(3) are derived at the percolation threshold[20] and therefore cannot account for the limiting cases of very high and very low saturation conditions.

However, for the range of expected soil saturation levels in natural soils, the theoretical relations fit the laboratory data very well.

## Numerical simulations of the two-phase reaction–diffusion equation in porous media

Based on the water–air spatial distribution from the invasion–percolation simulations, finite-element simulations of the diffusion–reaction equation were performed[22].

Within the system, two phases are considered, the (air) invading phase, $\Phi_a$ and the (water) depending phase, $\Phi_w$. Chemical components (i.e., $O_2$, $CO_2$ and dissolved organic carbon, $C_s$) are transported solely by diffusion, where each component has a phase-dependent diffusion coefficient. Surface reactions (Michaelis–Menten kinetics) take place on the perimeter of solid grains[58] in $\Phi_w$, with rate

$$R_{surf} = V_m \frac{C_s}{C_s + K_{m(s)}} \frac{C_{O_2}}{C_{O_2} + K_{m(O_2)}}, \qquad (4)$$

where $V_m$ is the maximum local rate of HR reaction [mol s$^{-1}$ m$^{-2}$], $C_{O_2}$ is the dissolved oxygen concentration, $C_s$ is the substrate concentration and $K_{m(s)}$ and $K_{m(O_2)}$ are the Michaelis constants of the substrate and oxygen, respectively. At the interface of the two phases, the mass transfer of chemicals is simulated by Henry's Law[59]. A fixed atmospheric concentration ($C0_{O_2}$) was set as the upper boundary condition for oxygen, while the other external boundaries were treated as open for all species. At the perimeter of the solid grains in $\Phi_w$ phase, a constant dissolved organic carbon concentration ($C0_{DOC}$) was set, to describe the organic matter (OM) degradation to dissolved organic carbon ($C_s$). Then, $C_s$ is transported by diffusion and reacts with dissolved oxygen, $C_{O_2}$ species, to produce dissolved $CO_2$ (Eq. (4)). 20 realizations were performed for each value of $\lambda_c$. The parameters used in the numerical simulations are given in Supplementary Table 2.

## Reaction–diffusion equation at the continuum scale for a single water patch

For a single water patch, the dissolved oxygen concentration ($C_{O_2}$) can be derived by solving the steady-state diffusion–reaction equation (DRE). Assuming that water patches can be approximated by a spherical shape, the DRE can be written as

$$D_m \left( \frac{d^2 C_{O_2}}{dr^2} + \frac{2}{r} \frac{dC_{O_2}}{dr} \right) = \frac{3\phi V_m}{\lambda_c} \frac{C_s}{C_s + K_{m(s)}} \frac{C_{O_2}}{C_{O_2} + K_{m(O_2)}}, \qquad (5)$$

where $D_m$ [m$^2$ s$^{-1}$] is the molecular diffusion coefficient of dissolved oxygen in water, $\phi$ is the medium porosity and $r$ is the radial distance. The term $\frac{3\phi}{\lambda_c}$ is the specific surface area [m$^2$ m$^{-3}$], obtained by assuming spherical grains as soil particles. By defining the following dimensionless parameters,

$$u = \frac{C_{O_2}}{C_0} \quad ; \chi = \frac{r}{r_0} \quad ; \alpha = \frac{K_{m(O_2)}}{C_0} \quad ; \beta = \frac{3V_m \phi C_s r_0^2}{D_m \lambda_c C_0 (C_s + K_{m(s)})}, \qquad (6)$$

we can rewrite Eq. (5) as

$$\frac{d^2 u}{d\chi^2} + \frac{2}{\chi} \frac{du}{d\chi} = \frac{\beta u}{u + \alpha}. \qquad (7)$$

Implementing boundary conditions $u(1) = 1$ and $u'(0) = 0$, Eq. (5) can be solved analytically for the limiting cases

$$u(\chi) = \begin{cases} \frac{1}{6}\beta(\chi^2 - 1) + 1; & \text{if } \alpha << u \\ \dfrac{e^{-\frac{\sqrt{\beta}(\chi-1)}{\sqrt{\alpha}}} \left( e^{\frac{2\sqrt{\beta}\chi}{\sqrt{\alpha}}} - 1 \right)}{\chi \left( e^{\frac{2\sqrt{\beta}}{\sqrt{\alpha}}} - 1 \right)}; & \text{if } \alpha >> u. \end{cases}$$

Otherwise, Eq. (7) can be solved numerically. To derive the characteristic reaction term, i.e., the right-hand side in Eq. (7), the solution for the normalized concentration, $u(\chi)$, is implemented in the reaction term. Then, the reaction rate depending on the radial coordinate can be obtained. Note, in the case of $\alpha << u$, the reaction term is equal to $\beta$ solely, without any radial dependency.

The analytical solutions for the limiting cases (where $\alpha << u$ or $\alpha >> u$) can be used to delineate the boundaries of the real solutions. As can be seen in Fig. 2b, at low $\beta$ values, the solutions can be restricted to a relatively narrow range (~ an order of magnitude) between the two analytical solutions (solid and dashed lines in Fig. 2b). This range increases as $\beta$ increases. Up to $\beta \approx 10$, the analytical and the full solution (solved numerically) show an excellent match. This suggests that analytical solutions for HR can be useful if $\beta$ is relatively small ($\beta < 10$). However, the value of $\beta$ is likely to vary strongly given the variation in the two main components that control soil HR: temperature and soil moisture (Supplementary Fig. 4a). Moreover, as we demonstrate in Fig. 2b, the assumption $\alpha >> u$ leads to a large difference with respect to the real solution when $\alpha$ is relatively small (the upper dashed line shows the analytical solution for $\alpha = 0.1$[60,61]. In contrast, for large $\alpha$ (the lower dashed line shows the analytical solution for $\alpha = 10$), the analytical solution captures the numerical solution. However, such higher $\alpha$ values have no meaning for aerobic respiration, where Michaelis constant, $K_m$, is always smaller than the saturated substrate concentration (i.e., it can be safely assumed that for aerobic microbes, the apparent $K_m$ for respiration is smaller than the maximum $O_2$ concentration observed at the soil-air interface).

In Supplementary Table 1, we summarize the parameters utilized for the percolation simulations and our HR model.

## Model sensitivity analysis

A sensitivity analysis of the model was performed using Monte Carlo simulations on daily data of soil temperature and moisture for the year 2021[31]. The simulation randomly sampled values from the daily dataset at each grid cell to evaluate the model input uncertainty, emerging from the monthly dataset used in this study. The results of the yearly heterotrophic respiration rate, as depicted in Supplementary Fig. 5a, exhibit a clear latitudinal dependence, even within 2000 independent realizations. The solid cloud in the figure represents the distribution of all realizations, while the dashed red line indicates the mean. The model demonstrates robustness and stability, even in the presence of random daily fluctuations in the data, with a relatively low deviation from the mean.

## Global databases and climate projections

To initialize the model parameters, we used a set of global gridded databases at a resolution of 0.25° and monthly time interval. Data came from remote sensing measurements (surface temperature, soil moisture, surface altitude and texture[31]), and by observation interpolation (dissolved organic carbon[33]).

Based on these data sets, for each grid point, we derived the following variables:

- Surface temperature
- Atmospheric oxygen concentration, based on the location's altitude and temperature[59]
- Dissolved oxygen concentration ($C_{O_2}$), according to Henry's law[59]
- Oxygen diffusion coefficient in water ($D_{O_2}$), based on the ambient temperature[62]
- Soil representative grain size ($\lambda_c$)[29]
- Soil surface area ($SSA$)[63]
- Soil saturation degree, and thus the resulting water patch characteristic size ($S_c$), and the number of patches ($N_c$).
- Maximum reaction rate, $V_m$, for Michaelis–Menten kinetics, based on the ambient temperature[58,60] and the soil surface area

- Dissolved organic carbon concentration[33].

Using these variables, Eqs. (1)–(3) were parameterized to each location on the global grid. In Supplementary Fig. 6, we present a schematic illustration of the model methodology for deriving heterotrophic respiration.

To predict future soil HR, we used climate projections from ten global circulation models from the Coupled Model Intercomparison Project Phase 6[41]. To maximize climate projection variability[64], we chose climate models that minimized the genealogy similarity between them[65]. We used the following models: ACCESS-CM2, EC-Earth3-Veg-LR, CNRM-CM6-1, NorESM2-MM, MPI-ESM1-2-LR, MIROC6, CanESM5, MIROC6, CMCC-CM2-SR5, CAMS-CSM1-0, and CESM2, and derived two variables: surface temperature and soil moisture in the upper portion of the soil column. The data were obtained with a monthly and 1° resolution from the worst-case greenhouse gas emission scenario (SSP5-8.5[66]) covering the period between 2015 (used as a reference year to represent present climate conditions) and 2100. Our model was then used to compute soil HR for each grid cell for the entire period. We present results of the simulation summarized for four climate zones: Tropical (0° to 23.5° N/S), Subtropical (23.5° to 40° N/S), Temperate (40° to 65° N/S), and Arctic (65° to 90° N/S). The evolution of soil HR for each of the climate models is presented in Supplementary Fig. 7.

To assess the potential changes in dissolved organic carbon (DOC) in the soil, as estimated by various Earth Systems Models[67], we applied upper and lower boundary constraints for the expected increase or decrease. We set an upper constraint of a 20% increase in the global DOC concentration in soils and a lower boundary of a 5% decrease[67]. The model was then run with a linear change until the end of the century based on the established upper and lower boundaries. As can be seen in Supplementary Fig. 5b, even under the upper and lower boundaries (which represent the extreme scenarios in DOC alteration), the model suggests only a minor change in the response of heterotrophic respiration. This demonstrates the importance of the availability of DOC for HR rather than its actual concentration and, thus, the role of soil moisture and temperature as the primary factors in controlling its availability for microorganism respiration. Additionally, the model suggests that microbial communities in the soil are able to efficiently utilize DOC, as indicated by the high local concentration of DOC compared to the Michaelis constant (Eq. (4)), and thus operate at full capacity (close to $V_m$) once DOC is available.

## Reporting summary

Further information on research design is available in the Nature Portfolio Reporting Summary linked to this article.

## Data availability

The data used to generate Figs 2-4, and Supplementary Fig. 3 can be accessed at the following link: https://gitlab.ethz.ch/anissan/global-warming-accelerates-soil-heterotrophic-respiration. Additionally, the raw data of the microfluidic experiments is available at https://zenodo.org/record/7918484#.ZFyxnOxBwrk. For more details and data, please contact the corresponding author (anissan@ethz.ch).

## Code availability

Computer codes are available online at: https://gitlab.ethz.ch/anissan/global-warming-accelerates-soil-heterotrophic-respiration.

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

## Acknowledgements

We thank Dr. Russell Naisbit for help with editing, and Maya Steindel for providing the illustration (Fig. 1). A.N. is supported by the ETH Postdoc Fellowship.

## Author contributions

A.N., M.H., N.P. and P.M. design the research. A.N., U.A. and N.G. designed and conducted experiments. A.N. developed and performed the numerical/theoretical simulations. A.N., U.A., N.P., N.G., J.J.M., P.M. and M.H. wrote the manuscript.

## Competing interests

The authors declare no competing interests.
