## [Peer Review File · Nature Communications]

Global warming accelerates soil heterotrophic respirationReviewer #1 (Remarks to the Author):

The study submitted by Nissan et al. is an attempt to calculate heterotrophic respiration flux at global scale and to perform future projections based on a fine scale mechanisms description. The approach is very original and the paper is well written. I am more a "global scale" guy so I am not sure I can evaluate deeply the fine scale equations but at least I think I understood.

Obviously I have some concerns but the authors may have ideas to solve them. In particular the evaluation/validation part needs to be improved.

First, there is no evaluation on Rh time series and since one of the aims of the paper is to predict future trajectories, I think an evaluation of the model on times series or at least on manipulative experiments on precipitation and/or temperature is necessary.

Secondly, I don't understand why on Extended Data Fig. 3 there only few points whereas the soil respiration database is much larger than that.

Finally, the upscaling from micro to global scale is done without validation at the field scale. To trust the upscaling procedure, it is absolutely necessary to show that the model predictions make sense at larger scales. Related to that, how the boundaries conditions are upscaled must be also better explain, in particular all the soil related boundaries conditions (SSA, grain sizes, Vm, DOC).

Another important missing point is how land use change (LUC) is considered. LUC is a major driver of soil C dynamic and therefore of Rh (Li et al., 2018; Wilson & Xenopoulos, 2009). In the manuscript it seems that LU is assumed to be fixed in the future but LUC is also part of the future scenarios and it must be taken into account or at least it must be discussed.

For the future projections, how changes in DOC are considered? DOC is a very dynamic pool and the DOC stocks will be affected by climate change (Bragazza et al., 2013; Pastor et al., 2003). If the authors assume a fix DOC for the future scenarios this is a very strong limitation.

I miss also direct comparisons between the Earth system model outputs (ESMs) and the model presented by the author for present day and for the future. The difference must be explained and if the two approaches behave the same then it is important to more clearly explain the added value of the approach proposed by this study compared to the ESMs.

Finally, since the readership of Nature Com. is not specialist of soil, I think that a table summarizing the name and the definition of the equations parameters and terms would be very useful for the reader. For instance, I am not sure I fully understood what water patches mean for the authors.

In eq. 4 the C O₂ is a bit misleading since it looks very similar with CO₂ and the reader may think you are presenting equations dealing with carbon dioxide.

In extended data Fig. 5 why there is not output arrows for ambient pressure?

References cited :

Bragazza, L., Parisod, J., Buttler, A., & Bardgett, R. D. (2013). Biogeochemical plant–soil microbe feedback in response to climate warming in peatlands. *Nature Climate Change*, 3(3), 273–277. <https://doi.org/10.1038/nclimate1781>

Li, W., Ciais, P., Guenet, B., Peng, S., Chang, J., Chaplot, V., Khudyaev, S., Peregon, A., Piao, S., Wang, Y., & Yue, C. (2018). Temporal response of soil organic carbon after grassland-related land-use change. In *Global Change Biology*. <https://doi.org/10.1111/gcb.14328>

Pastor, J., Solin, J., Bridgham, S. D., Updegraff, K., Harth, C., & Weishampel, P. (2003). Global warming and the export of dissolved organic carbon from boreal peatlands. *Oikos*, 100, 380–386.

Wilson, H. F., & Xenopoulos, M. A. (2009). Effects of agricultural land use on the composition of

fluvial dissolved organic matter. *Nature Geoscience*, 2(1), 37–41.
<https://doi.org/10.1038/ngeo391>

Reviewer #2 (Remarks to the Author):

The manuscript by Alon Nissan and colleagues submitted to *Nature Communications* provides a pore-scale mechanism to predict soil heterotrophic respiration, and this model was used to make predictions of soil heterotrophic fluxes for larger spatial and temporal scales, the results showed that the model yields estimates of recent trends in soil heterotrophic respiration rates at the global scale that are in line with observations. Lastly, this mechanistic model was used to simulate how soil heterotrophic respiration might change under global warming. It is glad to see that a pole-scale mechanisms model for heterotrophic respiration model was developed, and this model perform well at macro scale and large temporal scale. However, I have two concerns about this topic: 1) innovation: the impact of climate on heterotrophic respiration has been intensively studied, and the results from this study are not new; 2) uncertainty: the model they developed has many hypothesis and parameters, which make the results arguable, therefore, it is difficult to judge the advantages of this model over the previous soil heterotrophic respiration mechanism models or empirical models. Other than that, I also have some minor comments, please see below.

Line 44-50: The topic of this manuscript is heterotrophic respiration, I suggest here the author could directly talk about the importance of heterotrophic respiration as well as its response to global climate change rather than talking about soil respiration.

Line 50: Here "Soil carbon fluxes" mean same thing as soil respiration? If yes, use soil respiration instead rather than use two different terms represent a same thing.

Line 58-59: "Roughly a quarter of atmospheric CO₂ originates from soils, which is five times more than anthropogenic CO₂ emissions"; a quarter of atmospheric CO₂ = 600-800 Pg C × 1/4 = 150 - 200 Pg C; five times anthropogenic CO₂ emissions = 10 Pg C × 5 = 50 Pg C. There is a mismatch between them!

Line 67-72: a citation needed at the end of this sentence. Reference [15] is a good one to cite here.

Line 297: "as supported by observations [41],," this sentence has an extra comma.

Figure 3: How do you get this shift point, by visual or by statistic method?

Figure 4: can you explain y-axis scale in the figure caption? What is 1, 1.5, and 2 stand for?

Another suggestion is keep all numbers in y-axis with one decimal, i.e., using 1.0, 1.5, and 2.0 in the y-axis of panel a.

Reviewer #3 (Remarks to the Author):

The authors present a novel mechanistic model for heterotrophic respiration (HR), based on percolation theory and a two-phase reaction–diffusion equation system for water patches in porous media. It is interesting and refreshing to see a new, theoretically founded model in this area after more data-driven studies in the recent past. It also is remarkable, that apparently the model achieves good and almost unbiased estimates of HR from forward parametrization and first principles. Yet, I found unsatisfying that the authors chose "point estimates" for the parameters in the literature (e.g. DOC) and did not perform an uncertainty analysis of the model results.

Related, I am not sure if the global map of HR is fully plausible. HR rate of >3000g m⁻² yr⁻¹ appear very high. I agree with the authors, that we do not need to assume steady state in a transient regime (in a steady state HR = NPP (nota bene: not GPP!)). But typically estimates of NPP in tropical forests are max. 2000g, which means a loss of 1kg C per year, consuming all carbon within decades which is not plausible. Indeed, HR and soil respiration estimates are lower in the literature (e.g. Luysaeert et al 2007, Cleveland et al. 2011). To have a better insight on this, it would be helpful to have Fig 3b not only in relative units.

Thus, my mains concerns revolve around the global application, in particular in the changing climate context. Applying such a pore-scale model to global scale can only be considered as "back-

of-envelope" estimate, thus hard to defend beyond an interesting hypothesis. For instance, with changing climate and CO₂ also C inputs into the soil change. Essentially the authors assume, that DOC flux to microbes is the rate limiting step, not any other process, which is an oversimplification. See e.g. publications by Lehmann et al., Kuzyakov et al. and Schmidt et al. But even if we stay within the "physical view" of the system, I believe, phase-transitions (freezing and thawing) would need to be considered in such a global model, to give another example.

On the contrary, I found the observation on 4b interesting, that for many systems microbes seem to operate on the "ridge" of $f(\theta, T)$ – I'd encourage analysing model properties in this direction, possibly in comparison with Moyano et al., who see interaction effects.

Cleveland, Cory C., Alan R. Townsend, Philip Taylor, Silvia Alvarez-Clare, Mercedes MC Bustamante, George Chuyong, Solomon Z. Dobrowski et al. "Relationships among net primary productivity, nutrients and climate in tropical rain forest: a pan-tropical analysis." *Ecology letters* 14, no. 9 (2011): 939-947.

Kuzyakov, Y. V., and Larionova, A. A.: Contribution of rhizomicrobial and root respiration to the CO₂ emission from soil (review), *Eurasian Soil Science*, 39, 753-764, 2006.

Lehmann, J., Hansel, C. M., Kaiser, C., Kleber, M., Maher, K., Manzoni, S., Nunan, N., Reichstein, M., Schimel, J. P., Torn, M. S., Wieder, W. R., and Kögel-Knabner, I.: Persistence of soil organic carbon caused by functional complexity, *Nature Geoscience*, 13, 529 - 534, 10.1038/s41561-020-0612-3, 2020.

Luyssaert, S., Inglima, I., Jung, M., Richardson, A. D., Reichstein, M., Papale, D., Piao, S. L., Schulze, E.-D., Wingate, L., Matteucci, G., Aragao, L., Aubinet, M., Beer, C., Bernhofer, C., Black, K. G., Bonal, D., Bonnefond, J.-M., Chambers, J., Ciais, P., Cook, B., Davis, K. J., Dolman, A. J., Gielen, B., Goulden, M., Grace, J., Granier, A., Grelle, A., Griffis, T., Grünwald, T., Guidolotti, G., Hanson, P. J., Harding, R., Hollinger, D. Y., Hutya, L. R., Kolari, P., Kruijt, B., Kutsch, W., Lagergren, F., Laurila, T., Law, B. E., Le Maire, G., Lindroth, A., Loustau, D., Malhi, Y., Mateus, J., Migliavacca, M., Misson, L., Montagnani, L., Moncrieff, J., Moors, E., Munger, J. W., Nikinmaa, E., Ollinger, S. V., Pita, G., Rebmann, C., Rouspard, O., Saigusa, N., Sanz, M. J., Seufert, G., Sierra, C., Smith, M.-L., Tang, J., Valentini, R., Vesala, T., and Janssens, I. A.: CO₂ balance of boreal, temperate, and tropical forests derived from a global database, *Global Change Biology*, 13, 2509-2537, 10.1111/j.1365-2486.2007.01439.x, 2007.

Moyano, F. E., Vasilyeva, N., Bouckaert, L., Cook, F., Craine, J., Yuste, J. C., Don, A., Epron, D., Formanek, P., Franzluebbers, A., Ilstedt, U., Katterer, T., Orchard, V., Reichstein, M., Rey, A., Ruamps, L., Subke, J. A., Thomsen, I. K., and Chenu, C.: The moisture response of soil heterotrophic respiration: interaction with soil properties, *Biogeosciences*, 9, 1173-1182, 10.5194/bg-9-1173-2012, 2012.

Schmidt, M. W. I., Torn, M. S., Abiven, S., Dittmar, T., Guggenberger, G., Janssens, I. A., Kleber, M., Koegel-Knabner, I., Lehmann, J., Manning, D. A. C., Nannipieri, P., Rasse, D. P., Weiner, S., and Trumbore, S. E.: Persistence of soil organic matter as an ecosystem property, *Nature*, 478, 49-56, 2011.

Dear Reviewers,

We thank you for the time spent reviewing our manuscript, “*Global warming accelerates soil heterotrophic respiration*”. We are pleased that you consider the work presented in this manuscript to be important and suitable for publication in *Nature Communications*, and we feel that your insightful comments and suggestions have improved the quality and understandability of our study and contributed to its lasting impact.

In the revised manuscript, we have thoroughly addressed all of the reviewers' concerns, comments, and suggestions point by point, as detailed below and we have adapted the manuscript accordingly. We have conducted additional simulations, including sensitivity analysis and the incorporation of various future scenarios of dissolved organic carbon (DOC) in the soil to enhance the robustness of our predictions. Furthermore, we have expanded our validation by including a time series of HR field observations and data from additional sites in the soil respiration database (SRDB). In addition, we discuss in detail the effect of land use, both on the terrestrial carbon cycle and on our results. We believe that these efforts have greatly strengthened our manuscript and framework, making it even more reliable and robust. The suggestions from the reviewers have thus greatly improved the quality of our work.

We have deposited the codes utilized in the model into a publicly accessible repository with a readme file containing instructions; the repository is at the following link:
<https://gitlab.ethz.ch/anissan/global-warming-accelerates-soil-heterotrophic-respiration/-/tree/main/>

REVIEWER COMMENTS

Reviewer 1

The study submitted by Nissan et al. is an attempt to calculate heterotrophic respiration flux at global scale and to perform future projections based on a fine scale mechanisms description. The approach is very original and the paper is well written. I am more a “global scale” guy so I am not sure I can evaluate deeply the fine scale equations but at least I think I understood.

We thank the reviewer for the attentive reading of our manuscript and the positive feedback.

Obviously I have some concerns but the authors may have ideas to solve them. In particular the evaluation/validation part needs to be improved.

Response: We have improved the evaluation and validation part as suggested by the reviewer and explained in detail below. In short, we have extended the model validation part, as can be seen in Extended Data Fig. 3 in the revised manuscript. In particular, we have included an additional validation based on field site time series observations (Extended Data Fig. 3c) of HR and more sites from the SRDB showing the match to mean behavior (Extended Data Fig. 3d) (see below).

First, there is no evaluation on Rh time series and since one of the aims of the paper is to predict future trajectories, I think an evaluation of the model on times series or at least on manipulative experiments on precipitation and/or temperature is necessary.

Thanks for the useful suggestion. In Extended Data Fig. 3c, we have validated our model against time series of heterotrophic respiration observations in a tropical forest (Wu, C., Liang, N., Sha, L. *et al.* Heterotrophic respiration does not acclimate to continuous warming in a subtropical forest. *Sci Rep*); using the reported temperature, moisture, and soil texture as the model input parameters. As can be seen in the figure, the model is in very good agreement with the experimental data, which substantiates the validity of our model and reliability for predicting future trajectories of heterotrophic respiration.

Secondly, I don't understand why on Extended Data Fig. 3 there only few points whereas the soil respiration database is much larger than that.

The reviewer correctly noted that we used only a part of the database across different latitudes for comparison to the model (this was done due to the lack of soil column length information for the majority of the data). The revised manuscript now includes the entire SRDB database (Extended Data Fig. 3d), which allows for a more comprehensive examination of the model. The line in the figure shows the best fit to the cloud of points (observed = 0.7 x predicted), indicating that the top soil layer in our model accounts for approximately 70% of the total observed CO₂ flux. This update improves the comparison and robustness of the model --- thank you!

Finally, the upscaling from micro to global scale is done without validation at the field scale. To trust the upscaling procedure, it is absolutely necessary to show that the model predictions make sense at larger scales. Related to that, how the boundaries conditions are upscaled must be also better explain, in particular all the soil related boundaries conditions (SSA, grain sizes, V_m, DOC).

We agree that it would not be prudent to trust the model predictions at a larger scale without proper validation, as this could potentially result in significant inaccuracies. In recognition of this, we have thoroughly validated our model through a range of observations, both in the laboratory and field conditions, and the entire soil respiration database. This validation is now better documented and supported by Extended Data Fig. 3, which presents the results of these tests. The validation at the global scale is shown in panel (d) of that Figure which now includes all points of the SRDB database. Our use of coarse-grained soil and climate conditions and variables of course results in increased uncertainties in the model. However, comparing the excellent simulation in time at a tropical forest site in Figure 3c with a RMSE of 149 gC m⁻² per yr to the RMSE of mean soil HR rates of 214 gC m⁻² per yr in Figure 3d across all climate zones shows that even at the field scale the performance is very good. As shown and discussed in the manuscript, aggregating the results still yields findings that align with observations. Utilizing bulk soil properties, such as SSA, DOC and grain size, is the only means of handling the inherent heterogeneity of natural soils, which exists at every scale. To obtain reasonable estimates, we must carefully choose an appropriate scale for our variables, as seen in our use of the finest possible resolution (0.25-degree) scale for global-scale analysis. With regards to the boundary conditions for upscaling of grain size and DOC, we do not upscale these parameters ourselves but take them directly from available global datasets, in which these parameters are provided at that resolution.

Another important missing point is how land use change (LUC) is considered. LUC is a major driver of soil C dynamic and therefore of Rh (Li et al., 2018; Wilson & Xenopoulos, 2009). In the

manuscript it seems that LU is assumed to be fixed in the future but LUC is also part of the future scenarios and it must be taken into account or at least it must be discussed.

Land use change can significantly impact carbon storage in soils and the associated HR flux, hence, the effects can also vary. On one hand, land use practices can enhance carbon sequestration and stabilize soil carbon, leading to positive feedback. On the other hand, certain land use activities such as deforestation or tillage can reduce soil carbon, resulting in negative feedback. However, the overall effect of land use on carbon storage is complex and difficult to predict at the global scale (as can be seen by the last IPCC report, Chapter 5), introducing additional uncertainties to the models predicting carbon storage and HR flux, rather than improving their predictability. Furthermore, the estimated contribution of LUC is approximately 0.1 Pg C per year, whereas heterotrophic respiration accounts for around 50-60 Pg C per year, as reported by the IPCC in 2021. This suggests that the difference in these estimates is more than two orders of magnitude. As a result, it can be inferred that the model predictions remain valid, even if LUC is not taken into consideration.

In the revised manuscript (lines: 331-342), we discussed the effects and importance of LU to the terrestrial carbon cycle as well as how the modelling framework could be extended to include LUC. A detailed quantitative analysis is however out of the scope of the present paper and left for future work.

For the future projections, how changes in DOC are considered? DOC is a very dynamic pool and the DOC stocks will be affected by climate change (Bragazza et al., 2013; Pastor et al., 2003). If the authors assume a fixed DOC for the future scenarios this is a very strong limitation.

For preparing the response to this important question we conducted additional simulations and analyses. In the revised manuscript, we examine the model results against future predictions of dissolved organic carbon content in a new figure in Extended Data Fig. 5b. Changes in DOC content were taken from:

Todd-Brown, K. E. O., Randerson, J. T., Hopkins, F., Arora, V., Hajima, T., Jones, C., Shevliakova, E., Tjiputra, J., Volodin, E., Wu, T., Zhang, Q., and Allison, S. D.: Changes in soil organic carbon storage predicted by Earth system models during the 21st century, Biogeosciences, 11, 2341–2356, <https://doi.org/10.5194/bg-11-2341-2014>, 2014.

Based on Todd-Brown et al., as an upper boundary for the change of DOC we used an increase of 20% with respect to the present DOC, while for the lower boundary, we used a decrease of 5%. As can be seen in the figure, the model produces similar results for mean soil HR flux under all cases, which demonstrates the importance of DOC availability given by the soil temperature and moisture, rather than the bulk concentration of DOC.

In the revised manuscript, we have included these additional simulations under three different DOC future scenarios (Extended Data Fig.5b), (i) decrease of 5%, (ii) constant, and (iii) increase of 20%; we discussed (lines: 321-330) and included (Extended Data Fig. 5b, and Methods, lines: 799-814).

I miss also direct comparisons between the Earth system model outputs (ESMs) and the model presented by the author for present day and for the future. The difference must be explained and if the two approaches behave the same then it is important to more clearly explain the added value of the approach proposed by this study compared to the ESMs.

In the revised manuscript, we have expanded our analysis by incorporating additional information regarding the future soil HR projections presented by 25 Earth System Models (Lynch, C., Hartin, C., Chen, M., and Bond-Lamberty, B.: *Causes of uncertainty in observed and projected heterotrophic respiration from Earth System Models, Biogeosciences.*). This review of ESMs shows significant disparities and uncertainties among the models, with most failing to accurately reproduce both past and present trends in HR from observations. This highlights the advantages of our approach, which not only offers a more intuitive understanding and greater flexibility in terms of input variable tuning, but also produces results that are in closer alignment with current HR observations. Furthermore, our approach predicts a more modest increase in HR levels by the end of the century, compared to the average prediction made by the 25 ESMs.

In the revised manuscript (lines 256-262) we added: “Recently, Lynch et al. carried out a comprehensive evaluation of 25 Earth System Models (ESMs) under the SSP5-8.5 scenario. The findings reported in their paper exhibit significant disparities and uncertainties among the models. The mean projection of HR by the end of the century is a 50% increase, while only a few ESMs successfully reproduce the historical HR. Our model provides a lower estimate of HR increase while retaining consistency with prior observations.”

Finally, since the readership of Nature Com. is not specialist of soil, I think that a table summarizing the name and the definition of the equations parameters and terms would be very useful for the reader. For instance, I am not sure I fully understood what water patches mean for the authors.

Done. In the revised manuscript, we have included a table (Extended Data. Table 1) that summarizes the parameters used in the model simulations and in the invasion-percolation simulations. "water patches" are defined as connected pores in the soil matrix filled with water, which can be considered as a standalone water volume and disconnected from other patches.

In eq. 4 the CO_2 is a bit misleading since it looks very similar with CO_2 and the reader may think you are presenting equations dealing with carbon dioxide.

This is a good observation. Through the new table (Extended Data Table 1), the readers can now easily understand each parameter in the model.

In extended data Fig. 5 why there is not output arrows for ambient pressure?

Thank you, in the revised manuscript we corrected this omission and include the missing arrow.

References cited :

Bragazza, L., Parisod, J., Buttler, A., & Bardgett, R. D. (2013). Biogeochemical plant–soil microbe feedback in response to climate warming in peatlands. *Nature Climate Change*, 3(3), 273–277. <https://doi.org/10.1038/nclimate1781>

Li, W., Ciais, P., Guenet, B., Peng, S., Chang, J., Chaplot, V., Khudyaev, S., Peregon, A., Piao, S., Wang, Y., & Yue, C. (2018). Temporal response of soil organic carbon after grassland-related land-use

change. In *Global Change Biology*. <https://doi.org/10.1111/gcb.14328>

Included.

Pastor, J., Solin, J., Bridgham, S. D., Updegraff, K., Harth, C., & Weishampel, P. (2003). Global warming and the export of dissolved organic carbon from boreal peatlands. *Oikos*, 100, 380–386.

Wilson, H. F., & Xenopoulos, M. A. (2009). Effects of agricultural land use on the composition of fluvial dissolved organic matter. *Nature Geoscience*, 2(1), 37–41. <https://doi.org/10.1038/ngeo391>

Included.

Reviewer 2

The manuscript by Alon Nissan and colleagues submitted to *Nature Communications* provides a pore-scale mechanism to predict soil heterotrophic respiration, and this model was used to make predictions of soil heterotrophic fluxes for larger spatial and temporal scales, the results showed that the model yields estimates of recent trends in soil heterotrophic respiration rates at the global scale that are in line with observations. Lastly, this mechanistic model was used to simulate how soil heterotrophic respiration might change under global warming. It is glad to see that a pore-scale mechanisms model for heterotrophic respiration model was developed, and this model perform well at macro scale and large temporal scale.

We appreciate the reviewer's positive feedback on our manuscript and have fully addressed her/his suggestions and comments as outlined below.

However, I have two concerns about this topic: 1) innovation: the impact of climate on heterotrophic respiration has been intensively studied, and the results from this study are not new;

The study of heterotrophic respiration (HR) has been extensively researched, and our contribution to this field is not the first attempt to estimate HR rates. However, our work represents a fundamentally new approach in this area and provides more robust estimates of future HR alteration. As far as we know, our study is the first to incorporate pore-scale details in the modeling process, and then upscale the processes to produce a global estimate of HR. Our modeling framework is both novel and practical, offering significant advantages over data-driven (machine learning) models or complex Earth System Models with many parameters – it requires few input parameters, all of which are physically based and reflect the main drivers of HR change, and it offers superior accuracy for a reduced computational cost compared to ESM models. Our approach has been validated through comparison with various observational data at different spatial scales, and demonstrates good agreement with these observations. Additionally, our results provide robust trends for past, present, and future HR. Overall, we believe that our approach represents a fresh perspective on modeling HR, as remarked by Reviewers 1 and 3, and offers a versatile tool for estimating its magnitude globally under climate change. It does not replace, but complements ESM models, and produces new insights.

the model they developed has many hypothesis and parameters, which make the results arguable, therefore, it is difficult to judge the advantages of this model over the previous soil heterotrophic respiration mechanism models or empirical models.

We agree that some parameters are required to describe the relevant processes, however, they are determined in a physically based manner from knowledge about the soil (grain size) and climatic forcing (temperature, soil moisture) parameters. This approach based on rigorous upscaling is very different from previous approaches that rely on empirical relations or ESM simulations, which we would argue actually have many input parameters that are difficult to constrain. In the revised manuscript, in response to the reviewer we have now included a sensitivity analysis to evaluate the impact of uncertainties in soil temperature and moisture on our predicted HR rates (lines 204-216). This analysis is presented in Extended Data Fig 5a, which demonstrates how changes in these factors can affect our results. Additionally, in response to a point by Reviewer #1, we have also included an analysis of the sensitivity of our model to changes in dissolved organic concentration in the soil (lines 321-330, 799-814), which is presented in Extended Data Fig. 5b. This sensitivity analysis highlights the crucial role that soil temperature and moisture play in determining the availability of DOC for microorganisms, providing valuable insights into the major environmental factors that impact our HR predictions.

Other than that, I also have some minor comments, please see below.

Line 44-50: The topic of this manuscript is heterotrophic respiration, I suggest here the author could directly talk about the importance of heterotrophic respiration as well as its response to global climate change rather than talking about soil respiration.

We are grateful for the reviewer's suggestion, but we believe that referring to the whole soil respiration in the opening paragraph of the introduction is important to framing the general context of the manuscript.

Line 50: Here "Soil carbon fluxes" mean same thing as soil respiration? If yes, use soil respiration instead rather than use two different terms represent a same thing.

In the revised manuscript (lines 45-53), we have modified the text to: "Within the terrestrial carbon cycle, soil respiration, the emission of CO₂ through root (autotrophic) and microbial (heterotrophic) respiration \cite{Kuzyakov2006}, is the largest carbon efflux into the atmosphere \cite{Raich1992,IPCC2021}. Therefore, reliable quantification of how soil respiration may be affected by climate change is critical for predicting future atmospheric CO₂ concentrations. However, estimating terrestrial carbon effluxes, primarily driven by soil respiration, is highly uncertain \cite{Li13104, Tharammal2019, Konings2019, Jian2022}. The global carbon budget is significantly impacted by terrestrial carbon fluxes, making it crucial to improve current estimates."

The term 'carbon fluxes' refers to the overall terrestrial carbon cycle; however, it has been revised to note that soil respiration primarily drives these fluxes.

Line 58-59: "Roughly a quarter of atmospheric CO₂ originates from soils, which is five times more than anthropogenic CO₂ emissions"; a quarter of atmospheric CO₂ = 600-800 Pg C × 1/4 = 150 - 200

Pg C; five times anthropogenic CO₂ emissions = 10 Pg C × 5 = 150 Pg C. There is a mismatch between them!

We thank the reviewer for his correction and in the revised manuscript, we have changed the text to read: “Roughly a fifth of atmospheric CO₂ originates from soils, which is ten times more than anthropogenic CO₂ emissions” (lines: 61-63).

67-72: a citation needed at the end of this sentence. Reference [15] is a good one to cite here.

Thank you for your comment. We have now added a citation at the end of the sentence in question, as suggested.

Line 297: “as supported by observations [41],,” this sentence has an extra comma.

Thank you for bringing this to our attention. We apologize for the error and have now removed the extra comma from the sentence in question.

Figure 3: How do you get this shift point, by visual or by statistic method?

In this case, the shifting point was estimated by visual inspection, and the slope is supported and validated by the Mann-Kendall test.

Figure 4: can you explain y-axis scale in the figure caption? What is 1, 1.5, and 2 stand for? Another suggestion is keep all numbers in y-axis with one decimal, i.e., using 1.0, 1.5, and 2.0 in the y-axis of panel a.

Thank you for your comment and suggestion. We apologize for the confusion and we updated the figure caption to provide a clear explanation of the y-axis scale.

Reviewer 3

The authors present a novel mechanistic model for heterotrophic respiration (HR), based on percolation theory and a two-phase reaction–diffusion equation system for water patches in porous media. It is interesting and refreshing to see a new, theoretically founded model in this area after more data-driven studies in the recent past. It also is remarkable, that apparently the model achieves good and almost unbiased estimates of HR from forward parametrization and first principles.

We appreciate the reviewer's positive feedback on our manuscript and have fully addressed her/his suggestions and comments as outlined below.

Yet, I found unsatisfying that the authors chose “point estimates” for the parameters in the literature (e.g. DOC) and did not perform an uncertainty analysis of the model results.

In the revised manuscript, we have now included a sensitivity analysis to evaluate the impact of uncertainties in soil temperature and moisture on HR rates in our model (lines 204-216, and Section 4.5 in the Methods). This analysis is presented in Extended Data Fig. 5a, which demonstrates how changes in these factors can affect our results. Additionally, we have also included an analysis of the sensitivity of our model to changes in dissolved organic concentration in the soil (lines 321-330, 799-814), which is presented in Extended Data Fig. 5b. This sensitivity analysis highlights the crucial role that soil temperature and moisture play in determining the availability of DOC for microorganisms, providing valuable insights into the major environmental factors that impact our HR predictions.

Related, I am not sure if the global map of HR is fully plausible. HR rate of $>3000\text{g m}^{-2}\text{ yr}^{-1}$ appear very high. I agree with the authors, that we do not need to assume steady state in a transient regime (in a steady state $\text{HR} = \text{NPP}$ (nota bene: not GPP!)). But typically estimates of NPP in tropical forests are max. 2000g , which means a loss of 1kg C per year, consuming all carbon within decades which is not plausible. Indeed, HR and soil respiration estimates are lower in the literature (e.g. Luysaeert et al 2007, Cleveland et al. 2011). To have a better insight on this, it would be helpful to have Fig 3b not only in relative units.

This is a good observation which could result from the reading of the original figure. The revised manuscript now includes an updated Fig. 3 with an additional panel demonstrating the HR distribution across different Koppen climate classifications. As shown in the figure, and in accordance with the reviewer's expectation, the highest HR rate was observed in tropical regions, with an average of approximately $1000\text{ gC m}^{-2}\text{yr}^{-1}$ and a peak rate of around $2000\text{ gC m}^{-2}\text{yr}^{-1}$.

Thus, my main concerns revolve around the global application, in particular in the changing climate context. Applying such a pore-scale model to global scale can only be considered as "back-of-envelope" estimate, thus hard to defend beyond an interesting hypothesis. For instance, with changing climate and CO_2 also C inputs into the soil change. Essentially the authors assume, that DOC flux to microbes is the rate limiting step, not any other process, which is an oversimplification. See e.g. publications by Lehmann et al., Kuzyakov et al. and Schmidt et al. But even if we stay within the "physical view" of the system, I believe, phase-transitions (freezing and thawing) would need to be considered in such a global model, to give another example.

We acknowledge the reviewer's point regarding the potential limitations of upscaling from the pore scale to the global scale. However, our results have demonstrated good agreement with various field-scale observations. Additionally, the pore-scale model has valuable benefits, such as its physically descriptive nature and the ability to make adjustments from a physical perspective. Regarding the reviewer's comment on the assumption of DOC flux being the rate-limiting step, our model takes into account both the limitations of oxygen (aerobic respiration) and DOC availability (by moisture and temperature) and its concentration in the soil matrix. As such, the rate-limiting step can vary at different sites and times due to the interplay of different parameters.

Regarding the phase transitions such as freezing and thawing, the reviewer is correct in pointing out that our current spatial and temporal resolution in the global grid may not be sufficient to capture these processes. However, our model is flexible and can optionally use a finer spatial and

temporal resolution to account for these changes, by adjusting the temperature, moisture, and DOC conditions, particularly in cold regions where phase transitions are more prevalent.

On the contrary, I found the observation on 4b interesting, that for many systems microbes seem to operate on the “ridge” of $f(\theta, T)$ – I’d encourage analysing model properties in this direction, possibly in comparison with Moyano et al., who see interaction effects.

We concur with the reviewer's assessment that the results presented in Fig. 4b, demonstrating the optimal temperature and moisture conditions under which microbes operate, are intriguing. Our team is actively pursuing further research in this area and using laboratory observations and has plans to publish follow-up papers on the topic.

Cleveland, Cory C., Alan R. Townsend, Philip Taylor, Silvia Alvarez-Clare, Mercedes MC Bustamante, George Chuyong, Solomon Z. Dobrowski et al. "Relationships among net primary productivity, nutrients and climate in tropical rain forest: a pan-tropical analysis." *Ecology letters* 14, no. 9 (2011): 939-947.

Kuzyakov, Y. V., and Larionova, A. A.: Contribution of rhizomicrobial and root respiration to the CO₂ emission from soil (review), *Eurasian Soil Science*, 39, 753-764, 2006.

Included.

Lehmann, J., Hansel, C. M., Kaiser, C., Kleber, M., Maher, K., Manzoni, S., Nunan, N., Reichstein, M., Schimel, J. P., Torn, M. S., Wieder, W. R., and Kögel-Knabner, I.: Persistence of soil organic carbon caused by functional complexity, *Nature Geoscience*, 13, 529 - 534, 10.1038/s41561-020-0612-3, 2020.

Luyssaert, S., Inglima, I., Jung, M., Richardson, A. D., Reichstein, M., Papale, D., Piao, S. L., Schulze, E.-D., Wingate, L., Matteucci, G., Aragao, L., Aubinet, M., Beer, C., Bernhofer, C., Black, K. G., Bonal, D., Bonnefond, J.-M., Chambers, J., Ciais, P., Cook, B., Davis, K. J., Dolman, A. J., Gielen, B., Goulden, M., Grace, J., Granier, A., Grelle, A., Griffis, T., Grünwald, T., Guidolotti, G., Hanson, P. J., Harding, R., Hollinger, D. Y., Hutyyra, L. R., Kolari, P., Kruijt, B., Kutsch, W., Lagergren, F., Laurila, T., Law, B. E., Le Maire, G., Lindroth, A., Loustau, D., Malhi, Y., Mateus, J., Migliavacca, M., Misson, L., Montagnani, L., Moncrieff, J., Moors, E., Munger, J. W., Nikinmaa, E., Ollinger, S. V., Pita, G., Rebmann, C., Rouspard, O., Saigusa, N., Sanz, M. J., Seufert, G., Sierra, C., Smith, M.-L., Tang, J., Valentini, R., Vesala, T., and Janssens, I. A.: CO₂ balance of boreal, temperate, and tropical forests derived from a global database, *Global Change Biology*, 13, 2509-2537, 10.1111/j.1365-2486.2007.01439.x, 2007.

Moyano, F. E., Vasilyeva, N., Bouckaert, L., Cook, F., Craine, J., Yuste, J. C., Don, A., Epron, D., Formanek, P., Franzluebbers, A., Ilstedt, U., Katterer, T., Orchard, V., Reichstein, M., Rey, A., Ruamps, L., Subke, J. A., Thomsen, I. K., and Chenu, C.: The moisture response of soil heterotrophic respiration: interaction with soil properties, *Biogeosciences*, 9, 1173-1182, 10.5194/bg-9-1173-2012, 2012.

Included.

Schmidt, M. W. I., Torn, M. S., Abiven, S., Dittmar, T., Guggenberger, G., Janssens, I. A., Kleber, M.,

Koegel-Knabner, I., Lehmann, J., Manning, D. A. C., Nannipieri, P., Rasse, D. P., Weiner, S., and Trumbore, S. E.: Persistence of soil organic matter as an ecosystem property, *Nature*, 478, 49-56, 2011.

Included.

Reviewer #1 (Remarks to the Author):

The revised version of the manuscript submitted by Nissan et al. addressed all my previous comments. I suggest to accept the manuscript for publication.

Reviewer #2 (Remarks to the Author):

The authors have thoroughly addressed all concerns, comments and suggestions provided by three reviewers. And I believe the quality of the manuscript have been improved. I do not have further questions on the main text, but I some questions about reproducibility of the analysis.

(1) In the Data availability and Code availability section: the authors mentioned that the data and code used can be accessed at the following link: <https://gitlab.ethz.ch/anissan/global-warming-accelerates-soil-heterotrophic-respiration>. I tried to get the data and code from the website, but it turns out that only ETH employees can sign up or login in the system? Anyway, I am not able to figure it out how to download the data and code. So my question is why not just share all the materials in a Github repository? It is much easier for users to download the data and code from Github.

(2) The authors also mentioned in the "Data availability" section that raw data of the microfluidic experiments is available at: Zenodo repository (link will be activated). My question is with those materials (raw data + data + code), can people reproduce all the results in the manuscript if they are interested in this topic?

(3) It seems that the authors will share the raw data if the manuscript is published, but if as a reviewer, if I want to try to reproduce the results during the reviewing process, will the authors provide all the code, data, and raw data needed to reproduce the results?

Reviewer #3 (Remarks to the Author):

I thank the authors for considering my comments. While I do like the modelling approach, I remain unconvinced of the validity of the application in the global and climate change context, because the model remains static, in the sense that carbon pools (e.g. DOC) are not state variables but rather fixed parameters. I do not see a theoretical justification for that and rather interpret the existing body of (theoretical and empirical) evidence (e.g. papers by Manzoni), that dynamic carbon pools are essential for modelling the HR. [As an aside: DOC seems to be at saturation, but it is not fully traceable: in Extended Data Table 2 the unit is mg L⁻¹ while Km is mol m⁻³, i.e. hard to compare]. In particular under climate change, a key questions is how carbon and nitrogen dynamics might override the pure thermodynamic effect of temperature by limiting substrate availability.

The validation of the model with data also remains quite weak. (Indeed I thought the log-scale of the old figure went to 3.5 which is 3160 g/yr, but acknowledge that the order of magnitude is correct). Nevertheless Ext Fig. 3 shows more than 2-3 fold deviation of the model from data (the slope line). [And it is not clear how the slope is calculated. Usually modelled needs to be on the x-axis, y-axis observed and then a normal linear regression be fitted.].

Given that temporal changes seem to be the key message of the paper, the single time-series in Ext Fig. 3c also is insufficient for validation.

Last but not least I am unconvinced of the sensitivity analysis regarding DOC. In Ext Table the DOC 1-sigma-uncertainty is given as 33%. In the sensitivity analysis it is -5% and +20% (why asymmetric?). Given the Michaelis-Menten saturation function, it is expected that positive uncertainties have a lower effect than negative ones. That said, it seems that a large uncertainty on the negative side will have also a large effect. Relative sensitivity $dy/y / dx/x$ seems to be around 1 ?

Reviewer #1 (Remarks to the Author):

The revised version of the manuscript submitted by Nissan et al. addressed all my previous comments. I suggest to accept the manuscript for publication.

Reviewer #2 (Remarks to the Author):

The authors have thoroughly addressed all concerns, comments and suggestions provided by three reviewers. And I believe the quality of the manuscript have been improved. I do not have further questions on the main text, but I some questions about reproducibility of the analysis.

Thank you!

(1) In the Data availability and Code availability section: the authors mentioned that the data and code used can be accessed at the following link: <https://gitlab.ethz.ch/anissan/global-warming-accelerates-soil-heterotrophic-respiration>. I tried to get the data and code from the website, but it turns out that only ETH employees can sign up or login in the system? Anyway, I am not able to figure it out how to download the data and code. So my question is why not just share all the materials in a Github repository? It is much easier for users to download the data and code from Github.

We have confirmed that the access link, <https://gitlab.ethz.ch/anissan/global-warming-accelerates-soil-heterotrophic-respiration>, is publicly accessible.

(2) The authors also mentioned in the “Data availability” section that raw data of the microfluidic experiments is available at: Zenodo repository (link will be activated). My question is with those materials (raw data + data + code), can people reproduce all the results in the manuscript if they are interested in this topic?

Of course; the raw data was deposited in Zenodo repository:
<https://zenodo.org/record/7918484#.ZFyxnOxBwrk>

(3) It seems that the authors will share the raw data if the manuscript is published, but if as a reviewer, if I want to try to reproduce the results during the reviewing process, will the authors provide all the code, data, and raw data needed to reproduce the results?

Rest assured that we will provide all the source codes that we used to generate the results of our study. It's worth mentioning that a considerable portion of the data we utilized is publicly available, such as NASA's GLDAS database; e.g., for Fig. 3. This means that anyone can download this data for free and use it to parameterize the model and reproduce our findings.

Reviewer 3 - I thank the authors for considering my comments. While I do like the modelling approach, I remain unconvinced of the validity of the application in the global and climate change context, because the model remains static, in the sense that carbon pools (e.g. DOC) are not state variables but rather fixed parameters.

I do not see a theoretical justification for that and rather interpret the existing body of (theoretical and empirical) evidence (e.g. papers by Manzoni), that dynamic carbon pools are essential for

modelling the HR. [As an aside: DOC seems to be at saturation, but it is not fully traceable: in Extended Data Table 2 the unit is mg L⁻¹ while Km is mol m⁻³, i.e. hard to compare]. In particular under climate change, a key question is how carbon and nitrogen dynamics might override the pure thermodynamic effect of temperature by limiting substrate availability.

We appreciate the reviewer's comment on the importance of dissolved organic carbon (DOC) as a dynamic and critical component of the soil matrix, affecting heterotrophic respiration (HR), along with other dynamic -environmental and -chemical factors such as nitrogen concentration, organic matter, pH, humidity, root activity, and microbial abundance and diversity (and more). While we are well aware and acknowledge the significance of these factors, here we build a model that captures the two basic mechanisms driving HR efflux with a robustness that current earth system models ESMS are not able to reach. And that is per se unique at this stage of knowledge and carbon efflux model development.

It is we hope clear to the reviewer that our model does not intend to replace those more comprehensive models that consider the entire carbon pools and dynamics (e.g., Zhang, X., Xie, Z., Ma, Z., Barron-Gafford, G. A., Scott, R. L., & Niu, G.-Y. (2022). *A microbial- explicit soil organic carbon decomposition model (MESDM): Development and testing at a semiarid grassland site. Journal of Advances in Modeling Earth Systems, 14*, e2021MS002485. <https://doi.org/10.1029/2021MS002485>), but propose a complementary and novel way to understand future soil CO₂ efflux and their controls. Compared to our model, ESMS models contain numerous "fitting" parameters that introduce additional uncertainty and may obscure causes and effects of carbon efflux increase by climate change. Our goal was to elucidate the main mechanisms that explain carbon efflux in a climate change context and for that, we simplified the approach while retaining the main mechanisms and drivers controlling HR.

Crucially, we want to highlight that the dynamics of DOC are not well understood at both the microscale and bulk scales, and thus more difficult to upscale for different ecosystems (see Guo, Z, Wang, Y, Wan, Z, et al. *Soil dissolved organic carbon in terrestrial ecosystems: Global budget, spatial distribution and controls. Global Ecol Biogeogr. 2020; 29: 2159– 2175. <https://doi.org/10.1111/qeb.13186>*). Therefore, we treated DOC as a static or quasi-static (as in the sensitivity analysis) variable, where the results align with laboratory and field observations of HR. In our sensitivity analysis of DOC we showed that varying DOC does not (significantly) affect our HR predictions (Ext. Data Fig. 5b), which is also consistent with ESMS that include dynamic DOC and nitrogen pools for future prediction of the fate of organic stocks on soils. Treating DOC as a dynamic pool in our model at this stage may lead to increased uncertainties instead of improving the accuracy of our results. DOC availability is influenced by several factors, including temperature, carbon source, vegetation type, water content, nutrients, and microbial communities, which are less well understood at the microscale level. However, we acknowledge that this is an important and active area of research, and we look forward to exploring this direction further in the future.

In our pore-scale numerical model, DOC is indeed assumed at saturation at the grain surface and comparison of our results with experimental observations (Ext. Data Fig. 3) showed that this is a

valid approach for many cases. In fact, our results suggest that DOC concentration, at the bulk in soils is often at saturated conditions concerning Michaelis- Menten kinetics property (This is based on the work by: *Langeveld, J., Bouwman, A.F., van Hoek, W.J. et al. Estimating dissolved carbon concentrations in global soils: a global database and model. SN Appl. Sci. 2, 1626 (2020).* <https://doi.org/10.1007/s42452-020-03290-0>), and its availability for microbes is primarily controlled by temperature and the way water is distributed in the soil medium. Although we acknowledge that our model neglects several processes, we are convinced that because it is based on traceable first-order principles and rigorous upscaling that accounts for the primary drivers controlling the magnitude of HR, it provides a useful complementarity to existing approaches and results – a fresh new look as one of the other reviewers stated.

We finally note that Referee 1 also had a concern about quasi static DOC and, after the inclusion of a sensitivity analysis in the paper, they were now satisfied with this.

In light of the reviewer's comments, we included an explanation to the discussion section (lines 312-329).

The validation of the model with data also remains quite weak. (Indeed I thought the log-scale of the old figure went to 3.5 which is 3160 g/yr, but acknowledge that the order of magnitude is correct). Nevertheless Ext Fig. 3 shows more than 2-3 fold deviation of the model from data (the slope line). [And it is not clear how the slope is calculated. Usually modelled needs to be on the x-axis, y-axis observed and then a normal linear regression be fitted.].

Despite the uncertainty in the soil respiration database (SRDB) and the parameterization inputs of our model based on the reanalysis global database, we found that the agreement between observations and predictions is satisfactory. In fact, given the low root mean square error (RMSE) it would qualify as excellent.

To clarify, the SRDB provides a single value of the heterotrophic respiration (HR) rate for each site over a specific time window. For example, at site X, the annual HR rate is Y for measurements taken between T1 to T2. We used global datasets to parameterize our model at a monthly interval, taking into account time-dependent variations, and solved HR per month for the entire range of observations. Then, we aggregated the results to provide an annual mean estimate. Therefore, although a few points show a 2-3 fold deviation, we consider the overall calibration to be excellent. The slope was calculated by fitting a linear trend.

We will make sure to make this clearer in the description of the figure in the revised manuscript.

Given that temporal changes seem to be the key message of the paper, the single time-series in Ext Fig. 3c also is insufficient for validation.

We appreciate the reviewer's recognition of the importance of time-dependent validation. However, we would like to emphasize that while Ext. Fig. 3c is the only plot that explicitly shows time on the x-axis, Fig. 4c also takes into account time-dependent conditions (as explained above). Furthermore, our global-scale predictions for both past and future scenarios are consistent with previous data-driven estimations (which are time-dependent too); e.g., *Lynch, C., Hartin, C., Chen, M., and Bond-Lamberty, B.: Causes of uncertainty in observed and projected heterotrophic respiration from Earth System Models, Biogeosciences Discuss. [preprint], <https://doi.org/10.5194/bg-2017-405>, in review, 2017.*

Last but not least I am unconvinced of the sensitivity analysis regarding DOC. In Ext Table the DOC 1-sigma-uncertainty is given as 33%. In the sensitivity analysis it is -5% and +20% (why asymmetric?). Given the Michaelis-Menten saturation function, it is expected that positive uncertainties have a lower effect than negative ones. That said, it seems that a large uncertainty on the negative side will have also a large effect. Relative sensitivity $dy/y / dx/x$ seems to be around 1 ?

To clarify, the uncertainty values presented in the external table only pertain to the results of the pore-scale simulations shown in Fig. 2, and do not reflect any uncertainty at a larger scale. The values of -5% and 20% were obtained from *Todd-Brown, K. E. O., Randerson, J. T., Hopkins, F., Arora, V., Hajima, T., Jones, C., Shevliakova, E., Tjiputra, J., Volodin, E., Wu, T., Zhang, Q., and Allison, S. D.: Changes in soil organic carbon storage predicted by Earth system models during the 21st century, Biogeosciences, 11, 2341–2356, <https://doi.org/10.5194/bg-11-2341-2014>, 2014 study and represent the lower and upper bounds, respectively. These values were chosen as end-members because they reflect the range of uncertainty in the literature. The asymmetry in the range arises from the fact that most Earth system models predict an increase in soil organic carbon storage in the future.*